# CHIRODIFF: MODELLING CHIROGRAPHIC DATA WITH DIFFUSION MODELS

**Ayan Das[1,2], Yongxin Yang[1,3], Timothy Hospedales[1,4,5], Tao Xiang[1,2] & Yi-Zhe Song[1,2]**

[1]SketchX, CVSSP, University of Surrey; [2]iFlyTek-Surrey Joint Research Centre on AI;
[3]Queen Mary University of London; [4]University of Edinburgh, [5]Samsung AI Centre, Cambridge
a.das@surrey.ac.uk, yongxin.yang@qmul.ac.uk, t.hospedales@ed.ac.uk,
{t.xiang, y.song}@surrey.ac.uk

## ABSTRACT

Generative modelling over continuous-time geometric constructs, a.k.a *chirographic data* such as handwriting, sketches, drawings etc., have been accomplished through autoregressive distributions. Such strictly-ordered discrete factorization however falls short of capturing key properties of chirographic data – it fails to build holistic understanding of the temporal concept due to one-way visibility (causality). Consequently, temporal data has been modelled as discrete token sequences of fixed sampling rate instead of capturing the true underlying concept. In this paper, we introduce a powerful model-class namely *Denoising Diffusion Probabilistic Models* or DDPMs for chirographic data that specifically addresses these flaws. Our model named "CHIRODIFF", being non-autoregressive, learns to capture holistic concepts and therefore remains resilient to higher temporal sampling rate up to a good extent. Moreover, we show that many important downstream utilities (e.g. conditional sampling, creative mixing) can be flexibly implemented using CHIRODIFF. We further show some unique use-cases like stochastic vectorization, de-noising/healing, abstraction are also possible with this model-class. We perform quantitative and qualitative evaluation of our framework on relevant datasets and found it to be better or on par with competing approaches.

## 1 INTRODUCTION

Chirographic data like handwriting, sketches, drawings etc. are ubiquitous in modern day digital contents, thanks to the widespread adoption of touch screen and other interactive devices (e.g. AR/VR sets). While supervised downstream tasks on such data like sketch-based image retrieval (SBIR) (Liu et al., 2020; Pang et al., 2019), semantic segmentation (Yang et al., 2021; Wang et al., 2020), classification (Yu et al., 2015; 2017) continue to flourish due to higher commercial demand, unsupervised generative modelling remains slightly under-explored. Recently however, with the advent of large-scale datasets, generative modelling of chirographic data started to gain traction. Specifically, models have been trained on generic doodles/drawings data (Ha & Eck, 2018), or more "specialized" entities like fonts (Lopes et al., 2019), diagrams (Gervais et al., 2020; Aksan et al., 2020), SVG Icons (Carlier et al., 2020) etc. Building unconditional neural generative models not only allows understanding the distribution of chirographic data but also enables further downstream tasks (e.g. segmentation, translation) by means of conditioning.

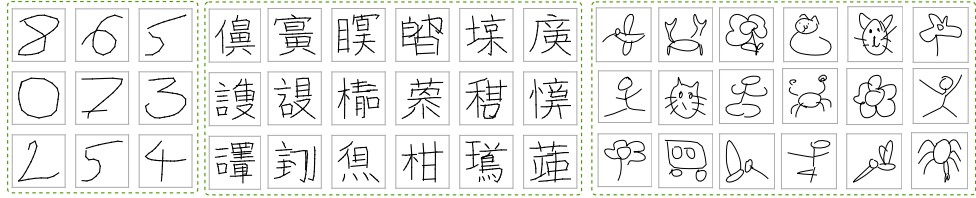

Figure 1: Unconditional samples from CHIRODIFF trained on VMNIST, KanjiVG and *Quick, Draw!*.

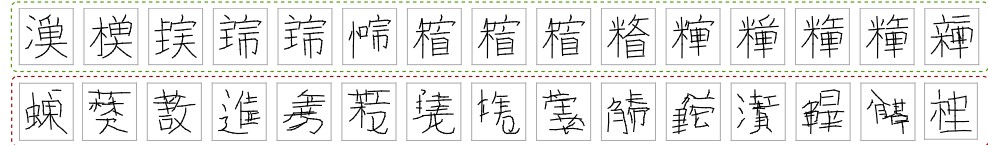

Figure 2: Latent space interpolation (Top) with CHIRODIFF using DDIM sampler and (Bottom) with auto-regressive model. CHIRODIFF's latent space is much more effective with compositional structures for complex data.

By far, learning neural models over continuous-time chirographic structures have been facilitated broadly by two different representations – grid-based raster image and vector graphics. Raster format, the de-facto representation for natural images, has served as an obvious choice for chirographic structures (Yu et al., 2015; 2017). The static nature of the representation however does not provide the means for modelling the underlying *creative process* that is inherent in drawing. "Creative models", powered by topology specific vector formats (Carlier et al., 2020; Aksan et al., 2020; Ha & Eck, 2018; Lopes et al., 2019; Das et al., 2022), on the other hand, are specifically motivated to mimic this dynamic creation process. They build distributions of a chirographic entity (e.g., a sketch) $X$ with a specific topology (drawing direction, stroke order etc), i.e. $p_\theta(X)$. Majority of the creative models are designed with autoregressive distributions (Ha & Eck, 2018; Aksan et al., 2020; Ribeiro et al., 2020). Such design choice is primarily due to vector formats having variable lengths, which is elegantly handled by autoregression. Doing so, however, restrict the model from gaining full visibility of the data and fails to build holistic understanding of the temporal concepts. A simple demonstration of its latent-space interpolation confirms this hypothesis (Figure 2). The other possibility is to drop the ordering/sequentiality of the points entirely and treat chirographic data as 2D point-sets and use prominent techniques from 3D point-cloud modelling (Luo & Hu, 2021a;b; Cai et al., 2020). However, point-set representation does not fit chirographic data well due to its inherently unstructured nature. In this paper, with CHIRODIFF, we find a sweet spot and propose a framework that uses non-autoregressive density while retaining its sequential nature.

Another factor in traditional neural chirographic models that limit the representation is effective handling of temporal resolution. Chirographic structures are inherently continuous-time entities as rightly noted by Das et al. (2022). Prior works like SketchRNN (Ha & Eck, 2018) modelled continuous-time chirographic data as discrete token sequence or motor program. Due to limited visibility, these models do not have means to accommodate different sampling rates and are therefore specialized to learn for one specific temporal resolution (seen during training), leading to the loss of spatial/temporal scalability essential for digital contents. Even though there have been attempts (Aksan et al., 2020; Das et al., 2020) to directly represent continuous-time entities with their underlying geometric parameters, most of them still possess some form of autoregression. Recently, SketchODE (Das et al., 2022) approached to solve this problem by using Neural ODE (abbreviated as NODE) (Chen et al., 2018) for representing time-derivative of continuous-time functions. However, the computationally restrictive nature of NODE's training algorithm makes it extremely hard to train and adopt beyond simple temporal structures. CHIRODIFF, having visibility of the entire sequence, is capable of implicitly modelling the sampling rate from data and consequently is robust to learning the continuous-time temporal concept that underlies the discrete motor program. In that regard, CHIRODIFF outperforms Das et al. (2022) significantly by adopting a model-class superior in terms of computational costs and representational power while training on similar data.

We chose *Denoising Diffusion Probabilstic Models* (abbr. as DDPMs) as the model class due to their spectacular ability to capture both diversity and fidelity (Ramesh et al., 2021; Nichol et al., 2022). Furthermore, Diffusion Models are gaining significant popularity and nearly replacing GANs in wide range of visual synthesis tasks due to their stable training dynamics and generation quality. A surprising majority of existing works on Diffusion Model is solely based or specialized to grid-based raster images, leaving important modalities like sequences behind. Even though there are some isolated works on modelling sequential data, but they have mostly been treated as fixed-length entities (Tashiro et al., 2021). Our proposed model, in that regard, is one of the first models to exhibit the potential to apply Diffusion Model on continuous-time entities. To this end, our generative model generates $X$ by transforming a discretized *brownian motion* with unit step size.

We consider learning stochastic generative model for continuous-time chirographic data both in unconditional (samples shown in Figure 1) and conditional manner. Unlike autoregressive models, CHIRODIFF offers a way to draw conditional samples from the model without an explicit encoder

when conditioned on homogeneous data (see section 5.4.2). Yet another similar but important application we consider is *stochastic vectorization*, i.e. sampling probable topological reconstruction $X$ given a perceptive input $\mathcal{R}(X)$ where $\mathcal{R}$ is a converter from vector representation to perceptive representation (e.g. raster image or point-cloud). We also learn deterministic mapping from noise to data with a variant of DDPM namely *Denoising Diffusion Implicit Model* or DDIM which allows latent space interpolations like Ha & Eck (2018) and Das et al. (2022). A peculiar property of CHIRODIFF allows a variant of the traditional interpolation which we term as "Creative Mixing", which do not require the model to be trained only on one end-points of the interpolation. We also show a number of unique use-cases like denoising/healing (Su et al., 2020; Luo & Hu, 2021a) and controlled abstraction (Muhammad et al., 2019; Das et al., 2022) in the context of chirographic data. As a drawback however, we loose some of the abilities of autoregressive models like stochastic completion etc.

In summary, we propose a Diffusion Model based framework, CHIRODIFF, specifically suited for modelling continuous-time chirographic data (section 4) which, so far, has predominantly been treated with autoregressive densities. Being non-autoregressive, CHIRODIFF is capable of capturing holistic temporal concepts, leading to better reconstruction and generation metrics (section 5.3). To this end, we propose the first diffusion model capable of handling temporally continuous data modality with variable length. We show a plethora of interesting and important downstream applications for chirographic data supported by CHIRODIFF (section 5.4).

## 2 RELATED WORK

Causal auto-regressive recurrent networks (Hochreiter & Schmidhuber, 1997; Cho et al., 2014) were considered to be a natural choice for sequential data modalities due to their inherent ability to encode ordering. It was the dominant tool for modelling natural language (NLP) (Bowman et al., 2016), video (Srivastava et al., 2015) and audio (van den Oord et al., 2016). Recently, due to the breakthroughs in NLP (Vaswani et al., 2017), interests have shifted towards non-autoregressive models even for other modalities (Girdhar et al., 2019; Huang et al., 2019). Continuous-time chirographic models also experienced a similar shift in model class from LSTMs (Graves, 2013) to Transformers (Ribeiro et al., 2020; Aksan et al., 2020) in terms of representation learning. Most of them however, still contains autoregressive generative components (e.g. causal transformers). Lately, *set* structures have also been experimented with (Carlier et al., 2020) for representing chirographic data as a collection of strokes. Due to difficulty with generating sets (Zaheer et al., 2017), their objective function requires explicit accounting for mis-alignments. CHIRODIFF finds a middle ground with the generative component being non-autoregressive while retaining the notion of order. A recent unpublished work (Luhman & Luhman, 2020) applied diffusion models out-of-the-box to handwriting generation, although it lacks right design choices, explanations and extensive experimentation.

Diffusion Models (DM), although existed for a while (Sohl-Dickstein et al., 2015), made a breakthrough recently in generative modelling (Ho et al., 2020; Dhariwal & Nichol, 2021; Ramesh et al., 2021; Nichol et al., 2022). They are arguably by now the de-facto model for broad class of image generation (Dhariwal & Nichol, 2021) due to their ability to achieve both fidelity and diversity. With consistent improvements like efficient samplers (Song et al., 2021a; Liu et al., 2022), latent-space diffusion (Rombach et al., 2021), classifier(-free) guidance (Ho & Salimans, 2022; Dhariwal & Nichol, 2021) these models are gaining traction in diverse set of vision-language (VL) problem. Even though DMs are generic in terms of theoretical formulation, very little focus have been given so far in non-image modalities (Lam et al., 2022; Hoogeboom et al., 2022; Xu et al., 2022).

## 3 DENOISING DIFFUSION PROBABILISTIC MODELS (DDPM)

DDPMs (Ho et al., 2020; Sohl-Dickstein et al., 2015) are parametric densities realized by a stochastic "reverse diffusion" process that transforms a predefined isotropic gaussian prior $p(X_T) = \mathcal{N}(X_T; \mathbf{0}, \mathbf{I})$ into model distribution $p_\theta(X_0)$ by de-noising it in $T$ discrete steps. The sequence of $T$ parametric de-noising distributions admit *markov property*, i.e. $p_\theta(X_{t-1}|X_{t:T}) = p_\theta(X_{t-1}|X_t)$ and can be chosen as gaussians (Sohl-Dickstein et al., 2015) as long as $T$ is large enough. With the model parameters defined as $\theta$, the de-noising conditionals have the form

$$p_\theta(X_{t-1}|X_t) := \mathcal{N}(X_{t-1}; \boldsymbol{\mu}_\theta(X_t, t), \boldsymbol{\Sigma}_\theta(X_t, t)) \qquad (1)$$

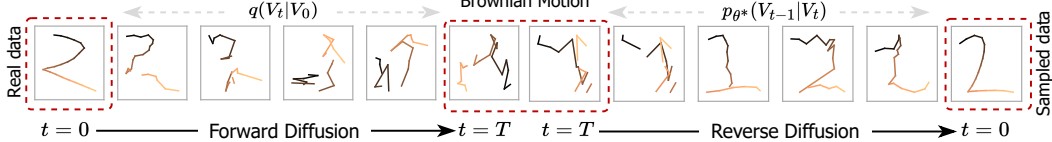

Figure 3: The forward and reverse diffusion on chirographic data. The "disconnected lines" effect is due to the pen-bits being diffused together. We show the topology by color map (black to yellow).

Sampling can be performed with a DDPM sampler (Ho et al., 2020) by starting at the prior $X_T \sim p(X_T)$ and running ancestral sampling till $t = 0$ using $p_{\theta^*}(X_{t-1}|X_t)$ where $\theta^*$ denotes a set of trained model parameters.

Due to the presence of latent variables $X_{1:T}$, it is difficult to directly optimize the log-likelihood of the model $p_\theta(X_0)$. Practical training is done by first simulating the latent variables by a given "forward diffusion" process which allows sampling $X_{1:T}$ by means of

$$q(X_t|X_0) = \mathcal{N}\left(X_t; \sqrt{\alpha_t}X_0; (1-\alpha_t)\mathbf{I}\right) \tag{2}$$

with $\alpha_t \in (0,1)$, a monotonically decreasing function of $t$, that completely specifies the forward noising process. By virtue of Eq. 2 and simplifying the reverse conditionals in Eq. 1 with $\mathbf{\Sigma}_\theta(X_t, t) := \sigma_t^2\mathbf{I}$, Sohl-Dickstein et al. (2015); Ho et al. (2020) derived an approximate variational bound $\mathcal{L}_{\text{simple}}(\theta)$ that works well in practice

$$\mathcal{L}_{\text{simple}}(\theta) = \mathbb{E}_{X_0 \sim q(X_0), t \sim \mathcal{U}[1,T], \, \boldsymbol{\epsilon} \sim \mathcal{N}(\mathbf{0}, \mathbf{I})}\left[||\boldsymbol{\epsilon} - \boldsymbol{\epsilon}_\theta(X_t(X_0, \boldsymbol{\epsilon}), t)||^2\right]$$

where a reparameterized Eq. 2 is used to compute a "noisy" version of $X_0$ as $X_t(X_0, \boldsymbol{\epsilon}) = \sqrt{\alpha_t}X_0 + \sqrt{1-\alpha_t}\boldsymbol{\epsilon}$. Also note that the original parameterization of $\boldsymbol{\mu}_\theta(X_t, t)$ is modified in favour of $\boldsymbol{\epsilon}_\theta(X_t, t)$, an estimator to predict the noise $\boldsymbol{\epsilon}$ given a noisy sample $X_t$ at any step $t$. Please note that they are related as $\boldsymbol{\mu}_\theta(X_t, t) = \frac{1}{\sqrt{1-\beta_t}}\left(X_t - \frac{\beta_t}{\sqrt{1-\alpha_t}}\boldsymbol{\epsilon}_\theta(X_t, t)\right)$ where $\beta_t \triangleq 1 - \frac{\alpha_t}{\alpha_{t-1}}$.

## 4 DIFFUSION MODEL FOR CHIROGRAPHIC DATA

Just like traditional approaches, we use the polyline sequence $X = \left[\cdots, \left(\mathbf{x}^{(j)}, p^{(j)}\right), \cdots\right]$ where the $j$-th point is $\mathbf{x}^{(j)} \in \mathbb{R}^2$ and $p^{(j)} \in \{-1, 1\}$ is a binary bit denoting the pen state, signaling an end of stroke. This representation is popularized by Ha & Eck (2018) and known as *Three-point format*. We employ the same pre-processing steps (equispaced resampling, spatial scaling etc) laid down by Ha & Eck (2018). Note that the cardinality of the sequence $|X|$ may vary for different samples.

CHIRODIFF is fairly similar to the standard DDPM described in section 3, with the sequence $X$ treated as a vector arranged by a particular topology. However, we found it beneficial not to directly use absolute points sequence $X$ but instead use velocities $V = \left[\cdots, \left(\mathbf{v}^{(j)}, p^{(j)}\right), \cdots\right]$, where $\mathbf{v}^{(j)} = \mathbf{x}^{(j+1)} - \mathbf{x}^{(j)}$ which can be readily computed using crude forward/backward differences. Upon generation, we can restore its original form by computing $\mathbf{x}^{(j)} = \sum_{j' \leq j} \mathbf{v}^{(j')}$. By modelling higher-order derivatives (velocity instead of position), the model focuses on high-level concepts rather than local temporal details (Ha & Eck, 2018; Das et al., 2022). We may use $X$ and $V$ interchangeably as they can be cheaply converted back and forth at any time.

Please note that we will use the subscript $t$ to denote the diffusion step and the superscript $(j)$ to denote elements in the sequence. Following section 3, we define CHIRODIFF, our primary chirographic generative model $p_\theta(V)$ also as DDPM. We use a forward diffusion process termed as "sequence-diffusion" that diffuses each element $(\mathbf{v}_0^{(j)}, p_0^{(j)})$ independently analogous to Eq. 2

$$q(V_t|V_0) = \prod_{j=1}^{|V|} q(\mathbf{v}_t^{(j)}|\mathbf{v}_0^{(j)})\prod_{j=1}^{|V|} q(p_t^{(j)}|p_0^{(j)}), \text{ with}$$

$$q(\mathbf{v}_t^{(j)}|\mathbf{v}_0^{(j)}) = \mathcal{N}(\mathbf{v}_t^{(j)}; \sqrt{\alpha_t}\mathbf{v}_0^{(j)}, (1-\alpha_t)\mathbf{I}), \quad q(p_t^{(j)}|p_0^{(j)}) = \mathcal{N}(p_t^{(j)}; \sqrt{\alpha_t}p_0^{(j)}, (1-\alpha_t)\mathbf{I})$$

Figure 4: The reverse process started with higher cardinality $|X|$ than the forward process.

Consequently, the prior at $t = T$ has the form $q(V_T) = \prod_{j=1}^{|V|} q(\mathbf{v}_T^{(j)}) \prod_{j=1}^{|V|} q(p_T^{(j)})$ where individual elements are standard normal, i.e. $q(\mathbf{v}_T^{(j)}) = q(p_T^{(j)}) = \mathcal{N}(\mathbf{0}, \mathbf{I})$. Note that we treat the binary pen state just like a continuous variable, an approach recently termed as "analog bits" by Chen et al. (2022). Our experimentation show that it works quite well in practice. While generating, we map the analog bit to its original discrete states $\{-1, 1\}$ by simple thresholding at $p = 0$.

With the *reverse sequence diffusion* process modelled as parametric conditional gaussian kernels similar to section 3, i.e. $p_\theta(V_{t-1}|V_t) := \mathcal{N}(V_{t-1}; \boldsymbol{\mu}_\theta(V_t, t), \sigma_t^2 \mathbf{I})$ and analogous change in parameterization (from $\boldsymbol{\mu}_\theta$ to $\boldsymbol{\epsilon}_\theta$), we can minimize the following loss

$$\mathcal{L}_{\text{simple}}(\theta) = \mathbb{E}_{V_0 \sim q(V_0),\ t \sim \mathcal{U}[1,T],\ \boldsymbol{\epsilon} \sim \mathcal{N}(\mathbf{0}, \mathbf{I})} \left[ ||\boldsymbol{\epsilon} - \boldsymbol{\epsilon}_\theta(V_t(V_0, \boldsymbol{\epsilon}), t)||^2 \right] \tag{3}$$

With a trained $\boldsymbol{\epsilon}_{\theta*}$, we can run DDPM sampler as $V_{t-1} \sim p_{\theta*}(V_{t-1}|V_t)$ (refer to section 3) iteratively for $t = T \to 1$. A deterministic variant, namely DDIM (Song et al., 2021a), can also be used as

$$V_{t-1} = \sqrt{\alpha_{t-1}} \left( \frac{V_t - \sqrt{1 - \alpha_t} \boldsymbol{\epsilon}_{\theta*}(V_t, t)}{\sqrt{\alpha_t}} \right) + \sqrt{1 - \alpha_{t-1}} \boldsymbol{\epsilon}_{\theta*}(V_t, t) \tag{4}$$

Unlike the usual choice of U-Net in pixel-based perception models (Ho et al., 2020; Song et al., 2021b; Ramesh et al., 2021; Nichol et al., 2022), CHIRODIFF requires a sequence encoder as $\boldsymbol{\epsilon}_\theta(V_t, t)$ in order to preserve and utilize the ordering of elements. We chose to encode each element in the sequence with the entire sequence as context, i.e. $\boldsymbol{\epsilon}_\theta(\mathbf{v}_t^{(j)}, V_t, t)$. Two prominent choices for such functional form are Bi-directional RNN (Bi-RNN) and Transformer encoder (Lee et al., 2019) with positional embedding. We noticed that Bi-RNN works quite well and provides much faster and better convergence. A design choice we found beneficial is to concatenate the absolute positions $X_t$ along with $V_t$ to the model, i.e. $\boldsymbol{\epsilon}_\theta(\cdot, [V_t; X_t], t)$, exposing the model to the absolute state of the noisy data at $t$ instead of drawing dynamics only. Since $X_t$ can be computed from $V_t$ itself, we drop $X_t$ from the function arguments now onward just for notation brevity. Please note that the generation process is non-causal as it has access to the entire sequence while diffusing. This gives rise to a non-autoregressive model and thereby focusing on holistic concepts instead of low-level motor program. This design allows the reverse diffusion (generation) process to correct *any* part of sequence from earlier mistakes, which is a not possible in auto-regressive models.

**Transforming "Brownian motion" into "Guided motion":** CHIRODIFF's generation process has an interesting interpretation. Recall that the reverse diffusion process begins at $V_T = [\cdots, (\mathbf{v}_T^{(j)}, p_T^{(j)}), \cdots]$ where each velocity element $\mathbf{v}_T^{(j)} \sim \mathcal{N}(\mathbf{0}, \mathbf{I})$. Due to our velocity-position encoding described above, the original chirographic structure is then $\mathbf{x}_T^{(j)} = \sum_{j'} \mathbf{v}_T^{(j')}$ which, by definition, is a discretized *brownian motion* with unit step size. With the reverse process unrolled in time, the brownian motion with full randomness transforms into a motion with structure, leading to realistic data samples. We illustrate the entire process in Figure 3.

**Length conditioned re-sampling:** A noticeable property of CHIRODIFF's generative process is that there is no *hard* conditioning on the cardinality of the sequence $|X|$ or $|V|$ due to our choice of the parametric model $\boldsymbol{\epsilon}_\theta(\cdot, t)$. As a result, we can kick-off the generation (reverse diffusion) process by sampling from a prior $p(V_T) = \prod_{j=1}^{L} q(\mathbf{v}_T^{(j)}) q(p_T^{(j)})$ of any length $L$, potentially higher than what the model was trained on. We hypothesize and empirically show (in section 5.3) that if trained to optimiality, the model indeed captures high level geometric concepts and can generate similar data with higher sampling rate (refer to Figure 4) with relatively less error. We credit this behaviour to the accessibility of the entire sequence $V_t$ (and additionally $X_t$) to the model $\boldsymbol{\epsilon}_\theta(\cdot)$. With the full sequence visible, the model can potentially build an internal (implicit) global representation which explains the resilience on increased temporal sampling resolution.

Figure 5: (A, B, C) Reconstruction CD against sampling rate factor. (D) Relative convergence time & sampling time (transparent bars) w.r.t our method. (E) FID of unconditional generation (averaged over multiple classes for QD).

# 5 EXPERIMENTS & RESULTS

## 5.1 DATASETS

**VectorMNIST or VMNIST (Das et al., 2022)** is a vector analog of traditional MNIST digits dataset. It contains 10K samples of 10 digits ('0' to '9') represented in polyline sequence format. We use 80-10-10 splits for our all our experimentation.

**KanjiVG**[1] is a vector dataset containing Kanji characters. We use a preprocessed version of the dataset[2] which converted the original SVGs into polyline sequences. This dataset is used in order to evaluate our method's effective on complex chirographic structures with higher number of strokes.

***Quick, Draw!*** **(Ha & Eck, 2018)** is the largest collection of free-hand doodling dataset with casual depictions of given concepts. This dataset is an ideal choice for evaluating a method's effectiveness on real noisy data since it was collected by means of large scale crowd-sourcing. In this paper, we use the following categories: {cat, crab, bus, mosquito, fish, yoga, flower}.

## 5.2 IMPLEMENTATION DETAILS

CHIRODIFF's forward process, just like traditional DDPMs, uses a linear noising schedule of $\left\{ \beta_{\min} = 10^{-4} \cdot 1000/T, \beta_{\max} = 2 \times 10^{-2} \cdot 1000/T \right\}$ as found out by Nichol & Dhariwal (2021); Dhariwal & Nichol (2021) to be quite robust. We noticed that there isn't much performance difference with different diffusion lengths, so we choose a standard value of $T = 1000$. The parametric noise estimator $\epsilon_\theta(\mathbf{v}_t^{(j)}, V_t, t)$ is chosen to be a bi-directional GRU encoder (Cho et al., 2014) where each element of the sequence is encoded while having contextual information from both direction of the sequence, making it non-causal. We use a 2-layer GRU with $D = 48$ hidden units for VM-NIST and 3-layer GRU for QuickDraw ($D = 128$) and KanjiVG ($D = 96$). We also experimented with transformers with positional encoding but failed to achieve reasonable results, concluding that positional encoding is not a good choice for representing continuous time. We trained all of our models by minimizing Eq. 3 using AdamW optimizer (Loshchilov & Hutter, 2019) and step-wise LR scheduling of $\gamma_e = 0.9997 \cdot \gamma_{e-1}$ at every epoch $e$ where $\gamma_0 = 6 \times 10^{-3}$. The diffusion time-step $t \in 1, 2, \cdots, T$ was made available to the model by concatenating sinusoidal positional embeddings (Vaswani et al., 2017) into each element of a sequence at every layer. We noticed the importance of reverse process variance $\sigma_t^2$ in terms of generation quality of our models. We found $\sigma_t^2 = 0.8\tilde{\beta}_t$ to work well in majority of the cases, where $\tilde{\beta}_t = \frac{1-\alpha_{t-1}}{1-\alpha_t}\beta_t$ as defined by Ho et al. (2020) to be true variance of the forward process posterior. Please refer to the project page[3] for full source code.

## 5.3 QUANTITATIVE EVALUATIONS

In order to assess the effectiveness of our model for chirographic data, we perform quantitative evaluations and compare with relevant approaches. We measure performance in terms of representation learning, generative modelling and computational efficiency. By choosing proper dimensions for competing methods/architectures, we ensured approximately same model capacity (# of parameters) for fare comparison.

---

[1]Original KanjiVG: kanjivg.tagaini.net

[2]Pre-processed KanjiVG: github.com/hardmaru/sketch-rnn-datasets/tree/master/kanji

[3]Our project page: https://ayandas.me/chirodiff

**Reconstruction** We construct a conditional variant of CHIRODIFF with an encoder $\mathcal{E}_V$ being a traditional Bi-GRU for fully encoding a given data sample $V$ into latent code $\mathbf{z}$. The decoder, in our case, is a diffusion model described in section 4. We sample from the conditional model $p_\theta(V_0|\mathbf{z} = \mathcal{E}_V(V))$ which is effectively same as standard DDPM but with the noise-estimator $\boldsymbol{\epsilon}_{\theta*}(V_t, t, \mathbf{z})$ additionally conditioned on $\mathbf{z}$. We expose the latent variable $\mathbf{z}$ to the noise-estimator by simply concatenating it with every ele-

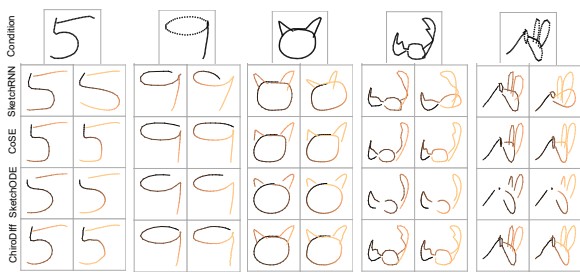

Figure 6: $1^{st}$ and $2^{nd}$ column for each example depicts sampling rate 1 & 2 respectively while reconstructing.

ment $j$ at all timestep $t \in [1, T]$. We also evaluate CHIRODIFF's ability to adopt to higher temporal resolution while sampling, proving our hypothesis that it captures concepts at a holistic level. We encode a sample with $\mathcal{E}_V$, and decode explicitly with a higher temporal sampling rate (refer to section 4). We compare our method with relevant frameworks like SketchODE (Das et al., 2022), SketchRNN (Ha & Eck, 2018) and CoSE (Aksan et al., 2020). Since autoregressive models like SketchRNN has no explicit way to increase temporal resolution, we train different models with re-sampled data, which is already disadvantageous. We quantitatively compare them with *Chamfer Distance (CD)* (Qi et al., 2017) (ignore the pen-up bit) for conditional reconstruction. Figure 5 shows the reconstruction CD against sampling rate factor (multiple of the original data cardinality) which shows the resilience of our model against higher sampling rate. SketchRNN being autoregressive, fails at high sampling rate (longer sequences), as visible in Figure 5(A, B & C). CoSE and SketchODE being naturally continuous, has a relatively flat curve. Also, we couldn't reasonably train SketchODE on the complex KanjiVG dataset (due to computational and convergance issues) and hence omitted from Figure 5. Qualitative examples of reconstruction shown in Fig. 6.

**Generation** We assess the generative performance of CHIRODIFF by sampling unconditionally (in 50 steps with DDIM sampler) and compute the FID score against the real data samples. Since the original inception network is not trained on chirographic data, we train our own on *Quick, Draw!* dataset (Ha & Eck, 2018) following the setup of (Ge et al., 2021). We compare our method with SketchRNN (Ha & Eck, 2018) and CoSE (Aksan et al., 2020) on all three datasets. Quantitative results in Figure 5(E) show a consistent superiority of our model in terms of generation FID. Quick-Draw FID values are averaged over the individual categories used. Qualitative samples are shown in Figure 1 and more in appendix A.1 with potential drawbacks highlighted. We also conducted a small scale user study to validate our generated samples and found our method to be superior (see appendix A.3 for details).

**Computational Efficiency** We also compare our method with competing methods in terms of ease of training and convergence. We found that our method, being from Diffusion Model family, enjoys easy training dynamics and relatively faster convergence (refer to Figure 5(D)). We also provide approximate sampling time for unconditional generation.

## 5.4 DOWNSTREAM APPLICATIONS

### 5.4.1 STOCHASTIC VECTORIZATION

An interesting use-case of generative chirographic model is *stochastic vectorization*, i.e. recreating plausible chirographic structure (with topology) from a given perceptive input. This application is intriguing due to human's innate ability to do the same with ease. The recent success of diffusion models in capturing distributions with high number of modes (Ramesh et al., 2021; Nichol et al., 2022) prompted us to use it for stochastic vectorization, a problem of inferring potentially multi-modal distribution. We simply condition our generative model on a perceptive input $\mathcal{X} = \mathcal{R}(X) = \{\mathbf{x}^{(j)}|(\mathbf{x}^{(j)}, p^{(j)}) \in X\}$, i.e. we convert the sequence into a point-set (also densely resample them as part of pre-processing). We employ a set-transformer encoder $\mathcal{E}_R(\cdot)$ with max-pooling aggregator (Lee et al., 2019) to obtain a latent vector $\mathbf{z}$ and condition the generative model similar to section 5.3, i.e. $p_\theta(V|\mathbf{z} = \mathcal{E}_R(\mathcal{X}))$. We evaluated the conditional generation with Chamfer Distance (CD) on test set and compare with Das et al. (2021) (refer to Figure 7).

### 5.4.2 IMPLICIT CONDITIONING

Unlike explicit conditioning mechanism (includes an encoder) described in section 5.3, CHIRODIFF allows a form of "Implicit Conditioning" which requires no explicit encoder. Such conditioning is more stochastic and may not be used for reconstruction, but can be used to sample similar data from a pre-trained model $p_{\theta^*}$. Given a condition $X_0^{\text{cond}}$ (or $V_0^{\text{cond}}$), we sample a noisy version at $t = T_c < T$ (a hyperparameter) using the forward diffusion (Eq. 2) process as $V_{T_c}^{\text{cond}} = \sqrt{\alpha_{T_c}}V_0^{\text{cond}} + \sqrt{1-\alpha_{T_c}}\epsilon$ where $\epsilon \sim \mathcal{N}(\mathbf{0}, \mathrm{I})$. We then utilize the trained model $p_{\theta^*}$ to gradually de-noise $V_{T_c}^{\text{cond}}$. We run the reverse process from $t = T_c$ till $t = 0$

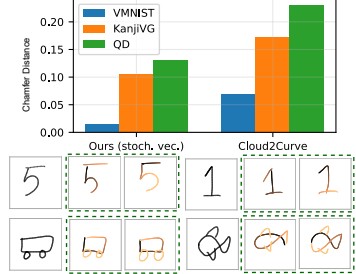

Figure 7: Stochastic vectorization. Note the different topologies (color map) of the samples.

$$V_{t-1} \sim p_{\theta^*}(V_{t-1}|V_t), \text{ for } T_c > t > 0, \text{ with } V_{T_c} := V_{T_c}^{\text{cond}}$$

The hyperparameter $T_c$ controls how much the generated samples correlate the given condition. By starting the reverse process at $t = T_c$ with the noisy condition, we essentially restrict the generation to be within a region of the data space that resembles the condition. Higher the value of $T_c$, the generated samples will resemble the condition more (refer to Figure 8). We also classified the generated samples for VMNIST & *Quick, Draw!* and found it to belong to the same class as the condition 93% of the time in average.

### 5.4.3 HEALING

The task of healing is more prominent in 3D point-cloud literature (Luo & Hu, 2021a). Even though the typical chirographic (autoregressive) models offer "stochastic completion" (Ha & Eck, 2018), it does not offer an easy way to "heal" a sample due to uni-directional generation. It is only very recently, works have emerged that propose tools for healing bad sketches (Su et al., 2020). With diffusion model family, it is fairly straightforward to solve this problem with CHIRODIFF. Given a "poor" chirographic data $\tilde{X}_0$, we would like to generate samples from a region in $p_{\theta^*}(X_0)$ close to $\tilde{X}_0$

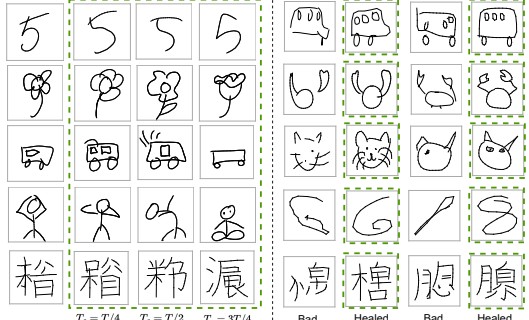

Figure 8: Implicit conditioning and healing.

in terms of semantic concept. Surprisingly, this problem can be solved with the "Implicit Conditioning" described in section 5.4.2. Instead of a real data as condition, we provide the poorly drawn data $\tilde{X}_0$ (equivalently $\tilde{V}_0$) as condition. Just as before, we run the reverse process starting at $t = T_h$ with $V_{T_h} := \tilde{V}_{T_h}$ in order to sample from a healed data distribution around $\tilde{V}_0$. $T_h$ is a similar hyperparameter that decides the trade-off between healing the given sample and drifting away from it in terms of high-level concept. Refer to Figure 8 (right) for qualitative samples of healing (with $T_h = T/5$).

### 5.4.4 CREATIVE MIXING

Creative Mixing is a chirographic task of merging two high-level concepts into one. This task is usually implemented as latent-space interpolation in traditional autoencoder-style generative models (Ha & Eck, 2018; Das et al., 2022). A variant of CHIRODIFF that uses DDIM sampler (Song et al., 2021a), can be used for similar interpolations. We use a pre-trained conditional model to decode the interpolated latent vector using $V_T = \mathbf{0}$ as a fixed point. Given two samples $V_{0_1}$ and $V_{0_2}$, we compute the interpolated latent variable as $\mathbf{z}_{\text{interp}} = (1 - \delta)\mathcal{E}_V(V_{0_1}) + \delta\mathcal{E}_V(V_{0_2})$ for any $\delta \in [0, 1]$ and run DDIM sampler shown in Eq. 4 with the noise-estimator $\epsilon_{\theta^*}(V_t, t, \mathbf{z}_{\text{interp}})$.

This solution works well for some datasets (KanjiVG & VMNIST; shown in Figure 9 (left)). For others (*Quick, Draw*), we instead propose a more general method inspired by ILVR (Choi et al., 2021) that allows "mixing" using the DDPM sampler itself. In fact, it allows us to perform mixing

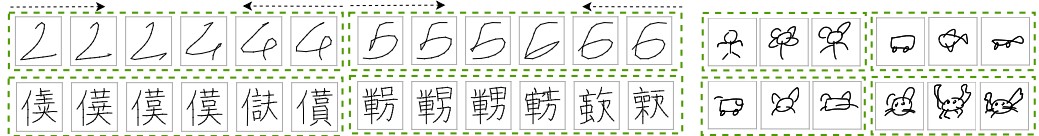

Figure 9: (Left) Latent-space semantic interpolation with DDIM. (Right) Creative Mixing shown as three-tuples consisting of $X_0$, $X_0^{\text{ref}}$ and the mixed sample respectively.

*without* one of the samples (reference sample) being known to the trained model. Given two samples of potentially different concepts, $X_0$ and $X_0^{\text{ref}}$ (or equivalently $V_0$ and $V_0^{\text{ref}}$), we sample from a pre-trained conditional model given $V_0$, but with a modified reverse process

$$X_{t-1} = X'_{t-1} - \Phi_\omega(X'_{t-1}) + \Phi_\omega(X_{t-1}^{\text{ref}})$$

$$\text{where, } V'_{t-1} \sim p_{\theta^*}(V_{t-1}|V_t, \mathbf{z} = \mathcal{E}_V(V_0), \text{ and } V_{t-1}^{\text{ref}} \sim q(V_{t-1}^{\text{ref}}|V_0^{\text{ref}})$$

where $\Phi_\omega(\cdot)$ is a *temporal* low-pass filter that reduces high-frequency details of the input data along *temporal axis*. We implement $\Phi_\omega(\cdot)$ using temporal 1D convolution of window size $\omega = 7$. Please note that for the operation in Eq. 5.4.4 to be valid, the sequences must be of same length. We simply resample the conditioning sequence to match the cardinality of the reverse process. The three-tuples presented in Figure 9(right) shows the mixing between different categories of *Quick, Draw!*.

### 5.4.5 Controlled Abstraction

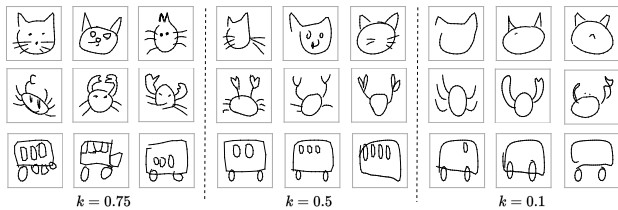

Figure 10: With reduced reverse process variance $\sigma_t^2 = k \cdot \tilde{\beta}$, generated samples looses high-frequency details but retains abstract concepts.

*Visual abstraction* is a relatively new task (Muhammad et al., 2019; Das et al., 2022) in chirographic literature that refers to deriving a new distribution (possibly with a control) that *holistically* matches the data distribution, but more "abstracted" in terms of details. Our definition to the problem matches that of Das et al. (2022), but with an advantage of being able to use the same model instead of re-training for different controls.

The solution of the problem lies in the sampling process of CHIRODIFF, which has an abstraction effect when the reverse process variance $\sigma_t^2$ is low. We define a continuous control $k \in [0, 1]$ as hyperparameter and use a reverse process variance of $\sigma_t^2 = k \cdot \tilde{\beta}_t$. The rationale behind this method is that when $k$ is near zero, the reverse process stops exploring the data distribution and converges near the *dominant modes*, which are data points conceptually resembling original data but more "canonical" representation (highly likely under data distribution). Figure 10 shows qualitative results of this observation on *Quick, Draw!*.

## 6 Conclusions, Limitations & Future Work

In this paper, we introduced CHIRODIFF, a non-autoregressive generative model for chirographic data. ChiroDiff is powered by DDPM and offers better holistic modelling of concepts that benefits many downstream tasks, some of which are not feasible with competing autoregressive models. One limitation of our approach is that vector representations are more susceptible to noise than raster images (see Figure 11 (Top)). Since we empirically set the reverse (generative) process variance $\sigma_t^2$, the noise sometimes overwhelms the model-predicted mean (see Figure 11 (Bottom)). Moreover, due the use of velocities, the *accumulated* absolute positions also accumulate the noise in proportion to

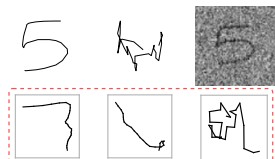

Figure 11: (Top) Noisy vector and raster data at same $\alpha$. (Bottom) Failures due to noise.

its cardinality. One possible solution is to modify the noising process to be adaptive to the generation or the data cardinality itself. We leave this as a potential future improvement on CHIRODIFF.

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

# A APPENDIX

## A.1 QUALITATIVE COMPARISONS

We make qualitative comparison of all the competing methods mentioned in this paper. We compare unconditionally generated samples for SketchRNN (Ha & Eck, 2018), CoSE (Aksan et al., 2020) and CHIRODIFF, with reconstructed samples for SketchODE (Das et al., 2022) since it is not primarily a generative model. The qualitative samples given below (figure 12, 13 & 14) for all three datasets from different models provide insights into their capabilities and drawbacks (we explicitly highlighted some of them).

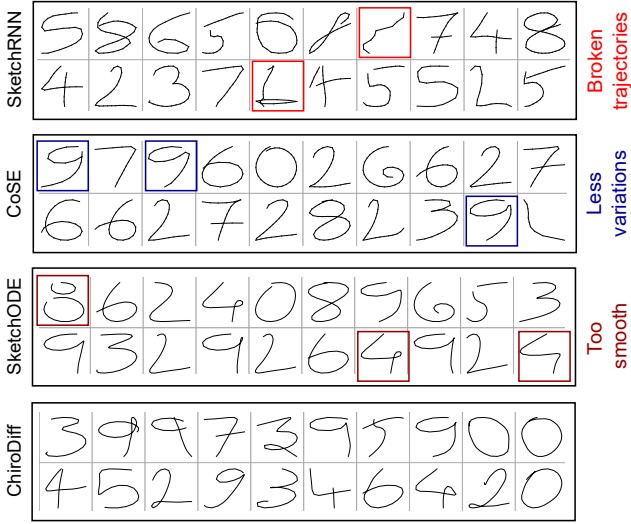

Figure 12: We show qualitative samples from different models trained on VectorMNIST dataset. We notice some common failure case in other models which is mitigated in CHIRODIFF. SketchRNN suffers from broken trajectories, CoSE suffers from less variability due to it's stroke-based generation and SketchODE produces overly smooth trajectories due to the truely continuous nature of the underlying representation.

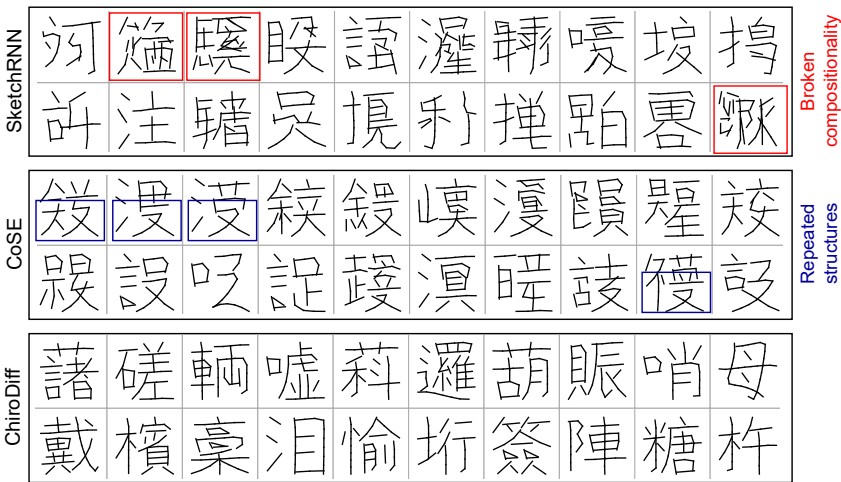

Figure 13: We show qualitative samples from different models trained on KanjiVG dataset. This dataset is particularly suitable for assessing model's capability to deal with large number of strokes and difficult compositionality. SketchRNN suffers from broken compositionality due to lack of holistic representation (causality), CoSE suffers from repeated stroke/structure in generation.

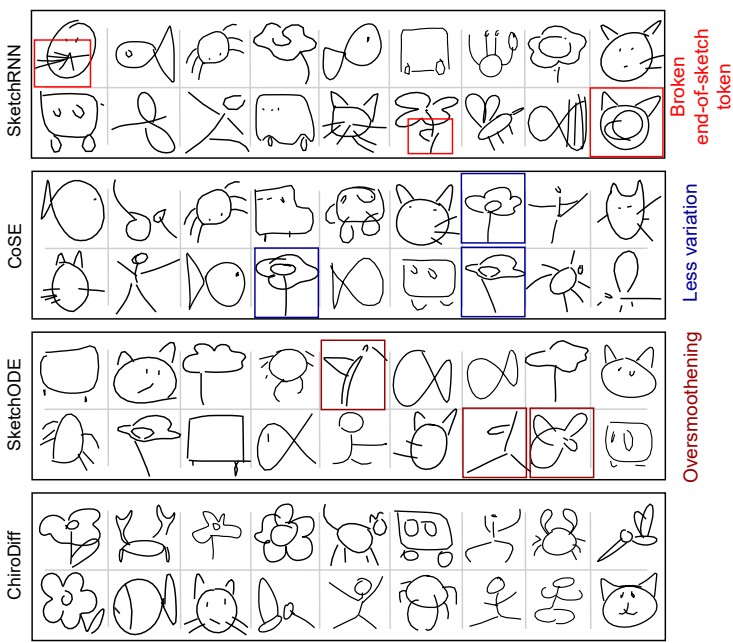

Figure 14: We show qualitative samples from different models trained on QuickDraw dataset. We noticed similar issues as mentioned for VectorMNIST dataset in Fig. 12.

## A.2   TABLE FOR QUANTITATIVE RESULTS

In this subsection, we provide the quantitative results of all of our experiments mentioned in the main paper in tabular format (table 1 & 2) in order to complement the graphical results of Fig. 5 & 7.

| Datasets | VMNIST | | | KanjiVG | | | Quick, Draw! | | |
|---|---|---|---|---|---|---|---|---|---|
| Scale-factor | 1.0 | 1.5 | 2.0 | 1.0 | 1.5 | 2.0 | 1.0 | 1.5 | 2.0 |
| SketchRNN (Ha & Eck, 2018) | 0.0123 | 0.0137 | 0.0190 | 0.163 | 0.237 | 0.413 | 0.163 | 0.227 | 0.373 |
| CoSE (Aksan et al., 2020) | 0.0213 | 0.0235 | 0.0280 | 0.143 | 0.172 | 0.235 | 0.133 | 0.172 | 0.244 |
| SketchODE (Das et al., 2022) | 0.0279 | 0.0311 | 0.0346 | – | – | – | 0.411 | 0.431 | 0.474 |
| **CHIRODIFF (ours)** | 0.0121 | 0.0135 | 0.0176 | 0.121 | 0.155 | 0.206 | 0.11 | 0.136 | 0.204 |

| Datasets | VMNIST | KanjiVG | Quick, Draw! |
|---|---|---|---|
| SketchRNN(Ha & Eck, 2018) | 1.31 | 2.7 | 1.3 |
| CoSE (Aksan et al., 2020) | 1.1 | 1.42 | 1.2 |
| SketchODE (Das et al., 2022) | 5.2 | – | 6.1 |
| **CHIRODIFF (ours)** | 1.0 | 1.0 | 1.0 |

| Datasets | VMNIST | KanjiVG | Quick, Draw! |
|---|---|---|---|
| SketchRNN(Ha & Eck, 2018) | 0.18 | 0.26 | 0.28 |
| CoSE (Aksan et al., 2020) | 0.31 | 0.40 | 0.46 |
| SketchODE (Das et al., 2022) | 0.63 | – | 0.86 |
| **CHIRODIFF (ours)** | 1.0 | 1.0 | 1.0 |

| Datasets | VMNIST | KanjiVG | Quick, Draw! |
|---|---|---|---|
| SketchRNN (Ha & Eck, 2018) | 13.11 | 32.12 | 39.71 |
| CoSE (Aksan et al., 2020) | 10.05 | 23.8 | 35.46 |
| **CHIRODIFF (ours)** | 10.57 | 15.31 | 25.12 |

Table 1: Three tables above provide the quantitative results complementing fig. 5. The first table shows reconstruction Chamfer Distance (CD) for various methods and datasets. The second and third tables depict the relative convergence and sampling time w.r.t ours. The third table shows unconditional generation FIDs.

| Datasets | VMNIST | KanjiVG | *Quick, Draw!* |
|---|---|---|---|
| Cloud2Curve (Das et al., 2021) | 0.07 | 0.106 | 0.230 |
| **CHIRODIFF (ours)** | 0.014 | 0.172 | 0.130 |

Table 2: The above table shows reconstruction CD between two methods comparing their stochastic vectorization capabilities. This table complements graphical results of Fig. 7.

## A.3 QUALITATIVE USER STUDY

We conduct a small scale user validation study in order to verify the effectiveness of our generative model. We first sampled a set of 20 samples from each model, i.e. CHIRODIFF, SketchRNN (Ha & Eck, 2018), CoSE (Aksan et al., 2020) and SketchODE as shown in appendix A.1. We show the set of generated samples along with 20 real samples side-by-side to 10 users independently and asked them to assess which one is *realistic*. Below tables shows the # of users identifying the generated samples to be real. As can be inferred from the table, samples generated by CHIRODIFF are labelled to be realistic by more users than competing methods.

| | VMNIST | KanjiVG | QuickDraw |
|---|---|---|---|
| SketchRNN (Ha & Eck, 2018) | 5 | 0 | 5 |
| CoSE (Aksan et al., 2020) | 6 | 3 | 4 |
| SketchODE (Das et al., 2022) | 2 | – | 0 |
| **CHIRODIFF (ours)** | 6 | 5 | 6 |

Table 3: Results of the user study. # of users labelled generated samples to be real.

