# OpenReview forum: "ChiroDiff: Modelling chirographic data with Diffusion Models"
_ICLR.cc/2023/Conference — ICLR 2023 poster_

### Official Review · Reviewer_VahX · 2022-10-23

**Confidence:** 4
**Correctness:** 3
**Technical Novelty And Significance:** 3
**Empirical Novelty And Significance:** 3
**Recommendation:** 6

**Clarity, Quality, Novelty And Reproducibility:**

The paper is well-motivated and well-written. The contribution claims are supported by empirical evidence. Though technical novelty is limited, the application of the diffusion models in this particular domain is novel and it introduces an interesting approach.

**Strength And Weaknesses:**

Strengths:

1- The paper applies a more powerful and expressive model class, the diffusion models, to chirographic data.

2- The paper is well-motivated around the holistic view achieved by the diffusion model. I think it is a novel approach in terms of how the sequential chirographic data is treated.

3- The proposed model has an interesting interpretation. At T, before any reverse diffusion steps, the chirographic structure is represented as Brownian motion. It basically treats every timestep as an independent data point by ignoring the causality. Nevertheless, it learns to transform this unstructured representation into realistic data samples. This introduces another interesting property that the model can handle sequences with arbitrary lengths and sampling frequencies (at least in theory; see the first weakness related to this).

---

Weaknesses:

1- One of the main claims of this work is the proposed model’s inherent advantage for modeling continuous-time data. I am not sure if this is entirely correct, as the backbone architecture is a Bi-RNN that processes the sequence step-by-step to approximate the noise. How could CHIRODIFF be better than the baselines like SketchODE or the decoder of CoSE? In particular, the latter can reconstruct a sequence at arbitrary resolution. Isn’t the underlying Bi-RNN prone to failure when the sequence length is arbitrarily increased (i.e., out-of-domain problem)?

2- In the stochastic vectorization task, the reconstructions are slightly different from the perceptive input. I am not sure if this is the expected result. While the model could capture different temporal modes and preserve semantics, it changes its appearance. Could it be due to the quality of the latent condition z or the stochasticity of the diffusion model-based decoder?

3- In contrast to the auto-regressive approaches, the proposed model lacks the ability to complete partial drawings. I think this is perfectly fine but it should be discussed in the paper.

4- This is not a weakness per se. Considering that the paper doesn’t introduce major modifications to the original diffusion model formulation, the background section on diffusion models seems a bit redundant. Some space could be saved for additional experiments, especially for ablation on the model’s performance on more complex data such as the hand-drawn diagram dataset of Gervais et al. or on computational efficiency (i.e., sampling-time vs. increasing sequence lengths, design choices, etc.).


**Summary Of The Paper:**

This paper proposes the use of diffusion models (DM), more specifically the Denoising Diffusion Probabilistic Models (DDPMs), for modeling chirographic data such as digital handwriting, sketching, and drawings. In contrast to the mainstream autoregressive approaches on sequential chirographic data, the proposed method, CHIRODIFF, processes the entire sequence in one go and hence better captures holistic concepts over sequences. The paper demonstrates the benefits of the proposed method in a variety of tasks, including de-noising, interpolation, and temporal reconstruction of images or 2D point clouds. CHIRODIFF is also compared with various architectures implementing different model classes, and it achieves better or competitive performance in reconstruction and generation tasks on the VectorMNIST, KanjiVG, and a subset of the QuickDraw datasets.

**Summary Of The Review:**

I find this paper useful for the community. It introduces a new model class and compares it against a carefully chosen set of baselines that implement different modeling concepts. Despite the advantages of the underlying diffusion models, there are also some weaknesses, which should be discussed in the rebuttal and incorporated into the paper. I am leaning toward accepting this paper if the authors provide clarifications in the rebuttal and address my concerns.

---------
**Post rebuttal update:** The rebuttal and the proposed revisions address my concerns. Hence, I am in favor of accepting this paper.

---

> ### Author Response · Authors · 2022-11-18
> **Author's response to reviewer #VahX**
>
> We thank the reviewer `#VahX` for agreeing on our motivation and acknowledging the novelty. Below we respond to the doubts put forward by the reviewer.
>
> 1. **Advantage for continuous data.** Firstly, we would like to clarify the claim of "inherently modelling continuous-time data" that it is more empirical than theoretical. We did not claim that the resolution agnostic nature is absolute and can be "arbitrarily increased" – we said, it ".. remains resilient to higher temporal sampling rate up to a good extent". It is also seen from the experiments (CD curves in Figure 5) that the error does indeed go up as resolution is increased, even for ChiroDiff. While it is not visually very different from CoSE (see Figure 5 and the new Figure 6), CoSE's error primarily comes from its prediction of the global geometric parameter. SketchODE, on the other hand, is extremely hard to properly fit for complex structures -- it has an over-smoothening problem (see also Figure 11 & 13 in supplementary A.1) and far from having a pixel-perfect reconstruction.
> That said, the reason we believe ChiroDiff possesses relative robustness against higher sampling rate is that its design is non-causal. Please note that every element of the sequence is reverse-diffused with the entire sequence as context, i.e. while proposing perturbation to the sequence as reverse diffusion unfolds, the model can "see" the holistic view of the chirographic structure. To make it even easier for the model, we explicitly make the "true" state of the structure $X$ available to the model along with its dynamics $V$ (see last paragraph of section 4). We hypothesize that this design helps the denoising model build an internal (implicit) global representation of the entire geometry, which makes it tolerant to increased sampling rate up to some extent. Input with higher granularity is indeed out-of-distribution for the RNN and for that reason, ChiroDiff's "local" reconstruction is slightly poor; but it's global reconstruction is better than others (see the 2nd columns of the new Figure 6).
>
> 2. **Stochastic vectorization.** You are right. We would like to clarify that the task we intended to solve is "plausible recreation" -- it is a cognitive task that resembles human's way of recreation which does incur information loss. There is another related task which is more computational, namely "vectorization" (studied mostly in computer graphics), which is a pixel-perfect vector reconstruction of a given perceptive input. There is nothing wrong with either of them. We chose to focus on the former. The result we show in (now) Figure 7 is expected, given that our model has a bottleneck (i.e. low-dimensional latent space z) while reconstructing. Any bottlenecked reconstruction pipeline is susceptible to information loss. An easy solution to mitigate this would be to increase the model capacity (parameters) or latent space dimension. Having said that, it is very much possible to create a conditional model without a bottleneck (using an attention mechanism) that can solve the pixel-perfect reconstruction problem. But then again, it would lose all the latent-space manipulation properties.
>
> 3. **Completion.** Even though we haven't considered this problem at all, it seems there is no straightforward way to do autoregressive-style completion. We have highlighted this in the paper now.
>
> 4. **Space Allocation.** On your and another reviewer's suggestion, we have trimmed down the section 3 and made room for some extra visualization, clarifications and equations. Please see the magenta colored text.
> Regarding your suggestion about using DiDi (Gervais et al.) dataset, even though there is in our method that limits application to this data, we feel this particular dataset is relatively better modelled with CoSE due to it’s repetitive nature of a small set of geometric structures (circles, rhombus, rectangle, arrows and some more). Moreover, we already use the KanjiVG dataset, which also focuses on high “compositionality” of strokes just like DiDi.
> Regarding computational efficiency, we have provided training convergence time in Figure 5. Sampling time is not really noticeably different for different sequence lengths with a good GPU for all practical purposes (length up to 200). There is however one parameter that affects the sampling time for ChiroDiff -- the diffusion steps $T$. We already mentioned that we use $T=1000$ as a "standard choice" and can be lowered much further (e.g. $T=50$) without noticeable difference in performance. We missed to mention that the unconditional generations are achieved by $T=50$ steps of DDIM sampler. We have now included a relative sampling time comparison in Figure 5.

---

> > ### Comment · Reviewer_VahX · 2022-11-20
> > **Thanks for the rebuttal**
> >
> > I've read other reviews and author responses. The authors' rebuttal and the proposed revisions address my concerns. Hence, I am in favor of accepting this paper. I have only the following suggestion:
> >
> > Wei Chen introduced prior work using diffusion models for handwriting modeling [1], which we were not aware of. I agree with the authors that their submission is technically different and aims for a larger scope. However, I think the claims of using the diffusion models for the first time are no longer valid. Even if [1] is not cited in the final version, the contribution claims should be corrected.
> >
> > [1] Luhman, Troy, and Eric Luhman. "Diffusion models for handwriting generation."

---

### Official Review · Reviewer_DeoY · 2022-10-24

**Confidence:** 4
**Correctness:** 4
**Technical Novelty And Significance:** 3
**Empirical Novelty And Significance:** 3
**Recommendation:** 6

**Clarity, Quality, Novelty And Reproducibility:**

Clarity: good.
Quality: good.
Novelty: It is a novel application of diffusion models.
Reproducibility: good.

**Strength And Weaknesses:**

Strength:
- To my best knowledge, this is the first work that explores diffusion models to generate temporal sequences.
- The proposed method overcomes the limitations of autoregressive models and achieves better empirical results on several datasets.

Weakness:
- The method directly adopts a previous diffusion model (DDPM) by replacing UNet with GRU. Thus, the technical novelty of the method is limited.

**Summary Of The Paper:**

This work proposed a method for generating online handwriting data based on diffusion models. On several datasets, the proposed method outperforms previous approaches.

**Summary Of The Review:**

Although the technical novelty of this work is limited, it is the first work that applies diffusion models to generating temporal data and achieves promising results on several datasets.

---

> ### Author Response · Authors · 2022-11-18
> **Author's response to reviewer #DeoY**
>
> We thank the reviewer `#DeoY` for acknowledging the novelty of the method and confirming it being the first work on diffusion models dealing with temporal sequences. We provide clarification regarding the weakness pointed out by the reviewer.
>
> We agree that this work is not targeted to be a fundamental modification on top of DDPM, nor have we ever claimed it. We were consistent in our claim that this work is a Diffusion Model ".. specifically suited for modelling continuous-time chirographic data", which hasn't received as much attention as pixel-image modelling. Our contribution, precisely, is to show the usage of non-autoregressive models in the context of sequential (chirographic) data. So far, the dominant belief was that arbitrary length sequence generative models must be causal in one way or the other [1, 2]. This was mostly due to the type of generative frameworks we had so far (e.g. VAEs). With the help of a new generative model-family, i.e. Diffusion Models, we can now lift the causality restriction. Moreover, our framework now enables many downstream tasks that were much harder to solve before, or weren't possible at all. Even though we focused on Chirographic data, the above mentioned principle could be used for any other type of sequential modality, which may encourage future works.
>
> The above points are clarified in the 2nd paragraph of the introduction section and also summarized in the last paragraph of the same section. Hope it helps.
>
>
> [1] David Ha and Douglas Eck. A neural representation of sketch drawings. In ICLR, 2018.
>
> [2] Emre Aksan, Thomas Deselaers, Andrea Tagliasacchi, and Otmar Hilliges. Cose: Compositional stroke embeddings. NeurIPS, 2020.

---

> > ### Comment · Reviewer_DeoY · 2022-11-27
> > **Thanks for response**
> >
> > I have carefully read the response letter and would like to maintain my rating.

---

### Official Review · Reviewer_N1HR · 2022-10-24

**Confidence:** 4
**Correctness:** 3
**Technical Novelty And Significance:** 3
**Empirical Novelty And Significance:** 4
**Recommendation:** 6

**Clarity, Quality, Novelty And Reproducibility:**

The clarity of this paper can be further improved by carefully revising the method description sections. The novelty is good. The authors are encouraged to release their code since it is not easy to reproduce the proposed method through reading those method descriptions and formulas.

**Strength And Weaknesses:**

Strength:
Recently, the superiority of diffusion models has been proven in the application of image synthesis. However, no work has been done to demonstrate the effectiveness of diffusion models in sequential data synthesis. This paper seems to be the first work to effectively using diffusion models to synthesize chirographic data. Therefore, I think many researchers will be interested in this paper, which makes some valuable academic contributions to the community. Despite some tiny representation problems, the paper is generally well-written and easy to follow.

Weaknesses:
1) The whole Section 3 provides a detailed description of DDPM, which is already well-known in the literature. I think this section should be shortened, removed, or integrated into Section 4 to emphasize the technical contribution of the proposed ChiroDiff.
2) Tables with accurate evaluation values should be provided instead of just displaying plots (Fig. 5 and 6). What is worse, no qualitative comparison has been made, and no user study has been conducted. In fact, for the tasks of image/graphic synthesis, qualitative evaluation is often more important than quantitative comparison. Many people want to directly compare the performance of two image/graphic synthesis methods by inspecting their synthesis results. Therefore, I suggest showing more qualitative results obtained by the proposed ChiroDiff and other existing approaches in the manuscript and its supplemental materials. Only in this manner, the advantage and necessity of modeling chirographic data via diffusion models can be clearly verified.


**Summary Of The Paper:**

This paper proposes a method to model chirographic data using diffusion models. Continuous-time geometric data (i.e., chirographic data) is typically modeled via autoregressive methods which fail to capture holistic information of the temporal data. On the contrary, the proposed ChiroDiff is capable of capturing holistic concepts and remaining resilient to higher temporal sampling rate. To my knowledge, this work is the first to model chirographic data with diffusion models. Both quantitative and qualitative experiments have been conducted to verify the effectiveness of the proposed method in several different applications.

**Summary Of The Review:**

The authors deserve the credit since they are the first to introduce diffusion models to handle the interesting task of chirographic data synthesis. However, there still exist some weaknesses in this paper as mentioned above. Otherwise, I could give it a higher score.

---

> ### Author Response · Authors · 2022-11-18
> **Author's response to reviewer #N1HR**
>
> We thank the reviewer `#N1HR` for recognizing the novelty of the paper and its potential contribution to the community. Below we respond to the weaknesses pointed out.
>
> 1. **DDPM description.** Thank you for the suggestion. We have now trimmed down parts of section 3 and made room for some extra clarifications, equations and qualitative results. Please see the texts highlighted with magenta.
> 2. **Additional details.** Firstly, we have now provided the numerical data complementing Figure 5 & 6 in the supplementary material (Appendix A.2). Secondly, we have added representative samples from different competing methods and different datasets (see new Figure 6 and appendix A.1). Moreover, we also highlighted some key failure modes for different methods which we observed. We also conducted a small scale user study and updated it in the paper (Appendix A.3).
>
> Regarding your concern about reproducibility, we will make our implementation public upon acceptance.

---

> > ### Comment · Reviewer_N1HR · 2022-11-29
> > **Thanks for your response**
> >
> > Thanks for the authors' response. I am positive on this paper. I think it is acceptable in ICLR after minor revision.

---

### Public Comment · ~wei_chen45 · 2022-11-13
**questions about the contribution**

In this paper, authors proposed that their method "is the first Model to exhibit the potential to apply diffusion model on continuous time entities." However, a very similar work Diffusion-Handwriting [1] has been proposed, using Diffusion models to generate continuous time series data.

I think there are too many similarities between this paper and Diffusion-handwriting. For example, DDPM is used to model the generation, and the data format is (x, y, pen state), and  can complete the task of generating handwriting.
I think one of the most primary difference between the two papers is that the generation of diffusion-handwriting is the fixed length, while ChiroDiff is variable length. This seems that the noise predicted network is RNN-based structure of ChiroDiff, while Diffusion-Handwriting uses the CNN-based structure. The continuous time series generation of RNN-based structure has been widely studied since Sketch-RNN [2], so ChiroDiff is more like a combination of sketch-RNN and diffusion-handwriting. The author seems to have exaggerated his contribution in the paper.

I expect authors to discuss the mainly contribution, or compare Diffusion-Handwriting in the paper.

[1] Luhman, Troy, and Eric Luhman. "Diffusion models for handwriting generation." arXiv preprint arXiv:2011.06704 (2020).
[2] Ha, David, and Douglas Eck. "A neural representation of sketch drawings." arXiv preprint arXiv:1704.03477 (2017).

---

> ### Author Response · Authors · 2022-11-18
> **Author's response to the public comment by Wei Chen**
>
> Thank you for taking interest in reading the paper and providing feedback, specifically highlighting the "Diffusion-Handwriting" article [1]. However, we respectfully disagree with your opinion and suggestion to a great extent. We put forward the following points.
>
> 1. Firstly, we weren't aware of [1] until very recently (after the submission deadline). It is mostly because [1] is only an arXiv submission and not peer-reviewed (we couldn't find any published version of [1]). Please note that the authors are NOT required to cite/compare arXiv-only submissions under ICLR’s policy. Please see the last question of the "FAQ for Reviewers" section in [Reviewer's Guide](https://iclr.cc/Conferences/2023/ReviewerGuide).
>
> 2. That said, we went through the article [1] and found it to be more like an out-of-the-box adoption of a diffusion model on a specific dataset (IAM handwriting) to solve a specific problem (content/style guided generation). Our submission, on the other hand, is much larger in scope -- handling arbitrary length sequences, several downstream applications, technical properties like non-causality, right choice of network and diverse datasets with varying complexity. In this regard, the Diffusion-Handwriting article [1] seems quite limited and incomplete as a paper since it clearly lacks technical discussion and experimentations. So, we do not feel obligated to compare with [1]. We may consider discussing/citing it *only* if the area chairs (AC) suggests.
>
> 3. Although we do agree that [1] does have an obvious similarity of using DDPM, that seems to be it. Our model-family (non-causal Bi-RNN) and the way of handling pen-bit are clearly different from [1]. From [1], we even found the technicalities around pen-bit modelling confusing and unclear. So the technical correctness of [1] is still questionable.
>
> 4. Your comment, ".. ChiroDiff is more like a combination of sketch-RNN [2] .." is quite a mis-characterization. Firstly, we never claimed credit for the $(\Delta x, \Delta y, p)$ data-format and SketcRNN [2] has been cited whenever appropriate. Secondly, characterizing our model as ".. combination of SketchRNN .." only because it contains an RNN, is not quite correct. You seem to have missed the point (as rightly acknowledged and re-iterated by review `#VahX`) that our model uses a bidirectional-RNN, which is a fundamentally different generative model-class w.r.t causality. We spent a significant part of the paper explaining the (non-)causality argument and showing its characteristics. SketchRNN uses an unidirectional RNN (autoregressive), which is a different generative model-class.
>
> Hope it helps.
>
> [1] Troy Luhman and Eric Luhman. Diffusion models for handwriting generation. CoRR,
> abs/2011.06704, 2020.
>
> [2] David Ha and Douglas Eck. A neural representation of sketch drawings. In ICLR, 2018.

---

### Author Response · Authors · 2022-11-18
**General comments from the authors**

Hello, all reviewers and ACs.

We heartily thank all reviewers for their constructive comments and unanimously recognizing the novelty of the paper. We made our best effort to clarify each reviewer individually. Here, we would like to make some general comment

1. We have now added an appendix (supplementary material) at the end of the main paper containing qualitative comparison results (Appendix A.1), complementary tables (Appendix A.2) and a small scale user study (Appendix A.3).
2. We will make our code public, upon acceptance.
3. We have also shortened section 3 as requested by multiple reviewers and made room for some qualitative results, clarifications and equations.

Thank you.

---

### Decision · Program_Chairs · 2023-01-20

**Decision:**

Accept: poster

**Justification For Why Not Higher Score:**

While the paper adds an interesting and useful point in the design space of sequence models for chirographic data, and it is likely to be of interest to many people + inspire some future work, there is not huge technical innovation here, nor does the paper present results that are so significant that they should be highlighted to the entire ICLR community.

**Justification For Why Not Lower Score:**

"Accept (poster)" is the lowest accept score, and all reviewers were in agreement that the paper ought to be accepted.

**Metareview: Summary, Strengths And Weaknesses:**

This paper uses a denoising diffusion probabilistic model (DDPM) to model chirographic data (i.e. sequential vector strokes, as in handwriting or sketching). In prior work, autoregressive models have dominated this space. DDPMs offer potential advantages, in particular avoid the problem of exposure bias / autoregressive drift.

Strengths: reviewers all generally agreed that this topic is timely and would likely be of interest to many people in the ICLR community. It could inspire future work in applying DDPMs to other kinds of mixed continuous/discrete sequential data. The paper is technically sound and well-executed.

Weaknesses: reviewers noted the lack of qualitative comparisons to prior work and the lack of a perceptual study. Some reviewers thought the paper lacked novelty, as it could be viewed as "just" a DDPM with the U-Net replaced with a GRU.

The initial scores for this paper were all on the accept side. The authors' response did a good job of answering reviewer concerns, and reviewers remained in favor of accepting the paper.

**Note From Pc:**

if the above contains the word "oral" or "spotlight" please see: "oral" presentation means -> notable-top-5% and "spotlight" means -> notable-top-25%. As stated in our emails, we are disassociating presentation type from AC recommendations